# Communication training and the prescribing pattern of antibiotic prescription in primary health care

Christoph Strumann[1]*, Jost Steinhaeuser[1], Timo Emcke[2], Andreas Sönnichsen[3], Katja Goetz[1]

**1** Institute of Family Medicine, University Hospital Schleswig-Holstein, Luebeck, Germany, **2** Association of Statutory Health Insurance Physicians of the Federal State of Schleswig-Holstein, Bad Segeberg, Germany, **3** Department of General Practice and Family Medicine, Medical University of Vienna, Center for Public Health, Vienna, Austria

* c.strumann@uni-luebeck.de

## Abstract

### Background

The treatment of upper respiratory tract infections (URTIs) accounts for the majority of antibiotic prescriptions in primary care, although an antibiotic therapy is rarely indicated. Non-clinical factors, such as time pressure and the perceived patient expectations are considered to be reasons for prescribing antibiotics in cases where they are not indicated. The improper use of antibiotics, however, can promote resistance and cause serious side effects. The aim of the study was to clarify whether the antibiotic prescription rate for infections of the upper respiratory tract can be lowered by means of a short (2 x 2.25h) communication training based on the MAAS-Global-D for primary care physicians.

### Methods

In total, 1554 primary care physicians were invited to participate in the study. The control group was formed from observational data. To estimate intervention effects we applied a combination of difference-in-difference (DiD) and statistical matching based on entropy balancing. We estimated a corresponding multi-level logistic regression model for the antibiotic prescribing decision of German primary care physicians for URTIs.

### Results

Univariate estimates detected an 11-percentage-point reduction of prescriptions for the intervention group after the training. For the control group, a reduction of 4.7% was detected. The difference between both groups in the difference between the periods was -6.5% and statistically significant. The estimated effects were nearly identical to the effects estimated for the multi-level logistic regression model with applied matching. Furthermore, for the treatment of young women, the impact of the training on the reduction of antibiotic prescription was significantly stronger.

**Data Availability Statement:** The dataset used and analyzed during the current study is extracted from administrative data and provided by the Association of Statutory Health Insurance

Physicians of the Federal State of Schleswig-Holstein. The data are secured by strict German data protection regulation and we are not allowed to share the data with other researchers. However, a request to ac-cess the data for scientific purposes and to replicate the findings can be made to the Associa-tion of Statutory Health Insurance Physicians (ASHIP) in official writing at Dr. Monika Schliffke, service@kvsh.de, ASHIP, Bismarckallee 1-6, 23795 Bad Segeberg. The relevant software codes and technical support to replicate the findings are stored in the Supporting Information files.

**Funding:** The study was funded by the German Federal Ministry of Health (grant-number: ZMVI1-2515NIK002). The funder of the study had no role in study design, data collection, data analysis, data interpretation, or writing of the report.

**Competing interests:** CS, KG, TE and JS: the authors declare that they have no competing interests. I have read the journal's policy and the author AS of this manuscript have the following competing interests: AS is an editor of the German version of EbMG. The online access is provided free of charge to the study PCPs by Verlagshaus der Aerzte, Vienna. This does not alter our adherence to PLOS ONE policies on sharing data and materials.

## Conclusions

Our results suggest that communication skills, implemented through a short communication training with the MAAS-Global-D-training, lead to a more prudent prescribing behavior of antibiotics for URTIs. Thereby, the MAAS-Global-D-training could not only avoid unneces-sary side effects but could also help reducing the emergence of drug resistant bacteria. As a consequence of our study we suggest that communication training based on the MAAS-Global-D should be applied in the postgraduate training scheme of primary care physicians.

## Introduction

The widespread use of antibiotics and the lack of new drug development serve as the main causes for the emergence of drug resistant bacteria [1], limiting the effectiveness of antimicro-bial therapy [2]. The rapid increase of resistant bacteria is regarded as one of the greatest threats to global health [3]. Infections with antibiotic-resistant bacteria may cause higher severity of illness, mortality rates, risk of complications, admissions to hospital, hospital length of stay and health care costs [2, 4–7].

Especially, not indicated antibiotic use is considered to be a primary cause of increasing risk of bacterial resistance [8]. Therefore, several initiatives address the improvement of pre-scribing practices of antibiotics worldwide [9–11]. A prominent example for an irrational use of antibiotics can be found in primary care, where primary care physicians (PCPs) often treat upper respiratory tract infections (URTIs) with antibiotics [12]. URTIs are one of the most common reasons for encounter in primary care and are mostly caused by viral infections, making antibiotic-therapy appropriate for only a small number of high risk patients [13]. However, the treatment of URTIs accounts for the majority of antibiotic prescriptions in pri-mary care [12, 14, 15], although there is very limited evidence for their benefits [16–18]. Besides characteristics of the physicians (e.g., specialty, training, experience), patients (e.g., sex, age, insurance status, comorbidities) and environmental factors (e.g. access to and quality of care), patient knowledge and expectations, as well as the physicians' assumptions regarding these expectations play a crucial role in the prescribing process [19–21]. Furthermore, evi-dence strongly suggests that antibiotic prescriptions are associated with a communication problem. Most patients seem to possess insufficient knowledge about the difference between viral and bacterial infections [22]. Due to the patients' belief that a previously received antibi-otic drug cured their infection, their expectations to receive antibiotic therapy when next pre-senting with URTI symptoms will increase [23]. Additionally, physicians may wrongly assume that the patient will demand antibiotics and preemptively prescribe the medicine [24–26]. Moreover, due to an overload of patients, physicians might not take the time to change the patient's expectations by explaining the differences between viruses and bacteria in an under-standable and effective way [27–29]. Therefore, patient expectations could strongly influence physicians, who are willing to prescribe an antibiotic to maintain a good relationship and to save time [20, 30, 31].

Communication trainings have been found to be effective in decreasing the antibiotic pre-scription rate [32–37]. Although the benefits of adequate communication skills are well known, they are not part of the postgraduate training scheme of any medical specialty in Ger-many [38]. In the Netherlands, a mandatory instrument for training and measuring physi-cians' communication and medical skills is widely used in under- and postgraduate training [39]. This instrument, named Maastricht history taking and advice scoring list (MAAS-Glo-bal), has been recently translated and adapted for use in Germany (MAAS-Global-D) [40].

The aim of this study was to investigate whether a communication training based on the MAAS-Global-D can reduce the rate of antibiotic prescribing for URTIs. Since the expectations of the patients and their perceptions of the physicians are subjective and might differ between patients, we additionally evaluate the intervention effect by the patient's age and sex to increase the insights of the communication effect.

## Materials and methods

### Data source

This study was based on the analysis of routine data of the years 2013 to 2016 from the Association of Statutory Health Insurance Physicians (ASHIP) of the federal state Schleswig-Holstein, located in Northern Germany. The ASHIP is in charge for the reimbursement of services that are provided to patients within the statutory health insurance system. The dataset covers 85% of the population and 83% of the PCPs of Schleswig-Holstein [41, 42]. The URTI cases were identified by the target-diagnoses of acute bronchitis, sinusitis and pharyngitis (classified by the International Classification of Diseases, version 10 (ICD-10) codes: J01.-; J02.-; J20.- [43]). We concentrate the analysis to these diagnoses, since only in some cases the use of antibiotics is suggested by respective guidelines within these diagnoses. For cases of acute bronchitis (J20) an antibiotic prescription is indicated for elderly patients as well as for those with a severe cardiac or respiratory disease or a congenital or acquired immunodeficiency [44]. In the case of acute pharyngitis (J02), the indications for an antibiotic therapy are: pharyngitis due to group A streptococcus bacterial infections (GAS pharyngitis), scarlet fever, peritonsillar abscess, a suspected serious illness or clinical worsening as well as consumptive diseases, immunosuppression and acute rheumatic fever in the personal or family history [45]. For acute sinusitis (J01), an antibiotic therapy should be considered for patients with specific risk factors, as well as complications such as severe headache, facial swelling, lethargy and acute exacerbation of recurrent sinusitis. Moreover, severe pain and an increased inflammation score complaints in the course of the disease and with fever above 38.5˚C [46].

Since the antibiotic prescriptions have been inferred based on the visit diagnoses, we excluded cases with additional diagnoses. This includes the presence of diagnoses regarding puerperium/pregnancy (O00-O99), further (bacterial) infections (A00 to A37, A39 to A79, J15, J17, J18) or chronic diseases (I50, J44, J45, C00 to C75). If the diagnosis had been made several times or more than one diagnosis had been made from the three groups (J01, J02, J20), the corresponding cases were also excluded. To increase the comparability of the included cases and, thus, minimize a potential estimation bias of the communication training effect, only cases of patients that were older than 18 years are included in the analysis.

### Recruitment and inclusion criteria

All primary care physicians in private practices, working in a contract with statutory public health insurance and with a work experience of at least five years, who have patients with at least one of the target-diagnoses between 2013 and 2015 were considered for the intervention. In total, 1554 (76%) primary care physicians of Schleswig-Holstein have been invited by letter to participate in a study named "Effects of communication training with the MAAS-Global-D on the prescription of antibiotics for respiratory infections".

### Study design and estimation strategy

The intervention and the previously planned randomized controlled trial (RCT) has been described by Hammersen et al. [47] (*Trial registration*: *DRKS00009566*). The study was

originally designed to consist of two interventional study arms. In addition to the communication training, the second intervention group received an educational introduction into the use of and online-access to *EbMG online* (Evidence–based Medicine Guidelines) [48]. This point-of-care online tool provides further information material on the prescribing of antibiotics for uncomplicated respiratory infections. Since the inclusion rate was lower than initially expected, both intervention groups have been consolidated. Furthermore, a comparison between the pooled intervention group and the control group did not yield significant results due to a lack of power because of the small sample size. Instead, we formed a control group from observational data and applied a combination of difference-in-difference (DiD) estimation and matching approach that is considered to reproduce the results of RCTs very well under certain assumptions [49]. For instance, under the assumption that the average outcomes for the intervention and control group would have followed parallel trends over time before intervention, the DiD estimator identifies causal effects by contrasting the change for the intervention and control groups in pre- and post-intervention outcomes [50]. However, the assumption of parallel trends might be implausible in our setting. For instance, if physicians recognized a too high antibiotic prescription rate for URTIs, they presumably tried reducing it. Therefore, they might have been more likely to respond to the training offer that advertised a reduction of the prescription rate through improved communication skills. Consequently, the evolutions of the prescription rates were suspected to differ between the intervention and control group if the control group, as in this case, had not been built upon a controlled randomization. An alternative identifying assumption is that the potential outcomes are independent of intervention status, conditional on past outcomes and covariates [51]. By means of balancing the intervention and control group according to pre-intervention outcomes and covariates all potential outcome trends are perfectly aligned and the DiD estimates can be interpreted as causal effects [49, 52]. However, recent studies showed that the combination of DiD and matching might also deliver biased estimates [53]. In order to enhance the robustness of our findings and minimize the risk of estimation bias we compared DiD estimates from both unmatched and matched (on pre- intervention outcomes) data [54].

In the search for relevant variables determining the decision to prescribe an antibiotic for a specific URTI case (our dependent variable) we first estimated a multi-level random effects logistic regression model based on case-, patient- and physician-level data of the pre-intervention period. A logistic regression model was chosen to account for the binary nature of the dependent variable. Moreover, the logit model showed computational merits and, unlike the probit model, it did not suffer from any convergence failures. Random effects were specified on the physician level to account for intra-physician variability [55]. In a second step, we aggregated the data on the physician level and matched the intervention and control groups according to aggregated pre- intervention outcomes and covariates by means of entropy balancing [56]. Based on the balanced data, in the third step, we estimated a multi-level random effects logistic DiD regression model using the weights of the physician-level from entropy balancing. Alternatively, we also specified fixed physician effects in the pre-intervention analysis and the DiD regression models. For all models the results between fixed and random effects models are very similar and we conclude there is no correlation between the explanatory variables and the individual effects. The physicians, who had previously been selected to the control group were excluded from the third step of the analysis, since we could not rule out that their prescribing behavior might have been affected by the cancellation of participation in the communication training.

The study was approved by the ethics committee of Luebeck University before the recruitment of participants on 9 June 2015 (number of approval: 15–139). Statistical analyses were performed with STATA 15 (StataCorp LLC, College Station, TX, USA).

## Intervention

The intervention group received a communication training with an interactive workshop character (two times 2.25 hours), which was held at the Institute of Family Medicine in February and March of 2016. It was delivered face-to-face by members of the research team, including an expert in physician-patient communication. The curriculum of the training was derived from the German version (MAAS-Global-D [40]) of the Dutch instrument MAAS-Global [39]. After establishing the relevance and success of physician-patient communication, the participant were provided with information concerning the associated evidence base regarding treatment of URTIs. Furthermore, they learned about the different communicative phases of a consultation, corresponding communication skills as well as general communication skills for the whole consultation (e.g. adequate provision of information, structuring and empathy, shared decision-making).

## Measurements

As the outcome variable we considered the binary choice whether an antibiotic was prescribed for a URTI case. The selection of potential determinants serving as control variables in both the pre- intervention and the DiD regression analyses was based on related previous literature [25, 57]. They can be classified into three categories: (i) case related (*year*, *quarter*, *diagnosis and its certainty*, *emergency service*), (ii) patient specific (*insurance status*, *age*, *sex*) and (iii) physician characteristics (*age*, *sex*, *number of URTI*-patients in that quarter). Seasonal effects and a general trend in the prescribing pattern were considered by respective dummy variables identifying the quarter and the year of the consultation, respectively. As the prescription rate might differ between the considered diagnoses, we introduced dummy variables for *sinusitis* and *pharyngitis* with *bronchitis* serving as reference. According to the German coding policy, primary care physicians are required to designate their diagnoses as validated (certain) or suspected (cases without an established definite diagnosis). We controlled for the cases with a *certain diagnosis* by including a respective dummy variable. Further, we distinguish whether the patient visited an emergency care center during the out-of-hours care (*emergency service*). Demographic variables of the patient were comprised of the *sex* (*sex* = 1: female), the *age* and the *insurance status* (normal, family or retired). The age was grouped by respective dummy variables for patients aged <35, 35–65, 65+ to allow for nonlinear age effects. In Germany, the insurance status signifies whether the patient is ordinary insured, retired or coinsured. Children and grandchildren aged below 25 as well as spouses that are unemployed, not self-employed and are not exceeding an income of EUR 450 per month are coinsured with an ordinary insurance member. The considered age and insurance status based clusters reflect different stages of life that might go along with different expectations about the treatment. At the physician level, we controlled for the specialty, since primary care physician workforce in Germany consists of general practitioners, physicians in general internal medicine and a declining number of practitioners without special training in primary care (12%). Previous studies have shown substantial differences in prescribing behavior between general internists and general practitioners [58]. Further, we considered the *age* and the *sex* of the physician. Finally, to approximate the workload of the physician's practice we included the number of total URTI patients in the respective quarter. The logarithmic function to this variable accounts for unequal variation.

In the intervention analysis, the *DiD* dummy variable identifies observations of the intervention group for the post-intervention period. To control for any other time-invariant differences between both groups a dummy variable *trained* is additionally included.

## Results

In the first part of the analysis (pre-intervention), the sample of the pre-intervention analysis (2013 to 2015) consisted of 315,752 adult patients with 476,260 cases from 2,189 PCPs. For the second part of the study, we invited 1,554 PCPs in SH to participate in the training. The group of interested participants has been divided randomly in a control and an intervention group with each 17 PCPs. Due to a lack of power, we alternatively form the control group from observational data. Do to so, we excluded the prior control group physicians ($n$ = 17) and practitioners without special training in primary care, since they are lacking in the intervention group ($n$ = 198). Moreover, 492 PCPs are not considered because they are not treating URTIs in each of the considered years, for instance since they are entering or leaving the ASHIP payment system during the study period. Finally, the intervention/control group in the intervention analysis consisted of 17/1,460 PCPs with 1,807/170,683 patients with 2,284/235,355 cases in the pre-intervention period (2013:q1 to 2015:q4) and 585/61,755 patients with 698/75,167 cases after the intervention (2016:q2 to 2016:q4) (Fig 1).

### Pre-intervention analysis

The mean values of the considered variables in the pre-intervention analysis and the regression results are shown in Table 1. An antibiotic was prescribed in half of the considered cases (49%).

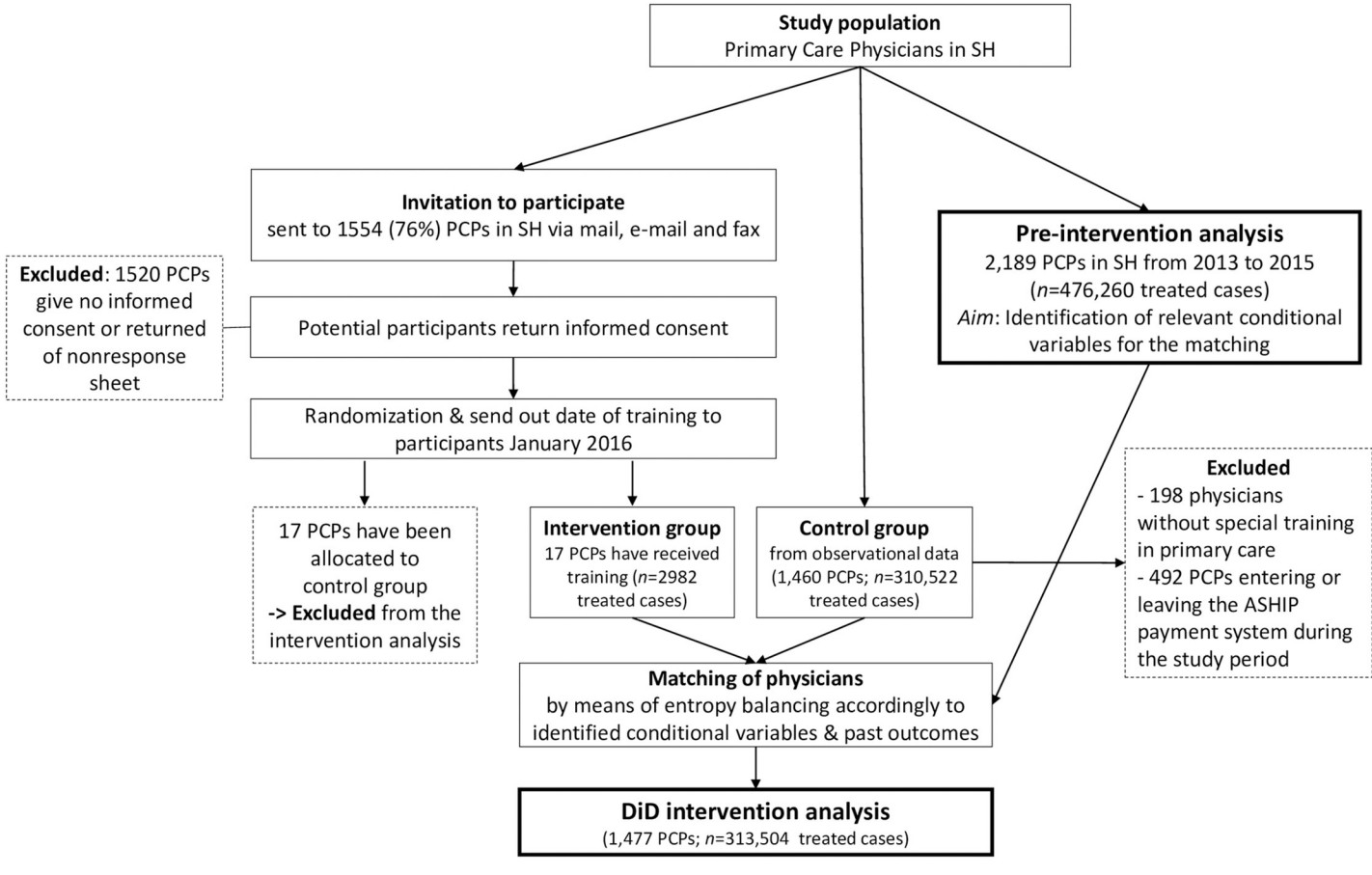

**Fig 1. Flow chart.**

**Table 1. Multilevel logistic regression analysis of the pre-intervention period of prescribing an antibiotic.**

| Variable | means | (1) | (2) |
|---|---|---|---|
| *Dependent variable* | | | |
| antibiotic prescription (= 1) | 0.49 | | |
| *Case characteristics* | | | |
| quarter | | | |
| 2nd quarter | 0.21 | -0.01 | -0.01 |
| 3rd quarter | 0.17 | -0.07** | -0.07** |
| 4th quarter | 0.26 | -0.08** | -0.08** |
| (Reference: 1st quarter) | | | |
| year | | | |
| 2014 | 0.32 | -0.03** | -0.03** |
| 2015 | 0.33 | -0.14** | -0.14** |
| (Reference: 2013) | | | |
| diagnosis | | | |
| sinusitis (J01) | 0.21 | -0.19** | -0.19** |
| pharyngitis (J02) | 0.29 | -0.18** | -0.18** |
| (Reference: bronchitis (J20)) | | | |
| certainty | | | |
| certain diagnosis | 0.99 | 0.38** | 0.38** |
| type of service | | | |
| emergency service | 0.03 | 0.40** | 0.40** |
| *Patient characteristics* | | | |
| Insurance status | | | |
| Family insured | 0.12 | 0.11** | 0.11** |
| Pensioners insured | 0.16 | 0.16** | 0.17** |
| (Reference: ordinary insured) | | | |
| Patient demographics | | | |
| Patient aged 35–65 | 0.52 | 0.33** | 0.45** |
| Patient aged 65+ | 0.14 | 0.28** | 0.39** |
| (Reference: < 35) | | | |
| Female patient | 0.59 | 0.09** | |
| (Reference: male) | | | |
| sex-age interactions | | | |
| Female patient aged <35 | 0.19 | | 0.22** |
| (Reference: male<35) | | | |
| Female patient aged 35–65 | 0.31 | | 0.02 |
| (Reference: male aged 35–65) | | | |
| Female patient aged 65+ | 0.09 | | 0.04 |
| (Reference: male aged 65+) | | | |
| *Physician characteristics* | | | |
| PCP specialty | | | |
| PCP without special training | 0.13 | 0.16 | 0.16 |
| General Internist | 0.18 | 0.00 | 0.00 |
| (Reference: GP) | | | |
| PCP demographics | | | |
| Physician age | 55.11 | 0.00 | 0.00 |
| Female physician | 0.31 | -0.04 | -0.04 |
| (Reference: male) | | | |

*(Continued)*

**Table 1.** (*Continued*)

| Variable | *means* | (1) | (2) |
|---|---|---|---|
| PCP workload | | | |
| *log*(#URTI-patients) | *3.60* | 0.15** | 0.15** |
| *intercept* | | -1.19** | -1.27** |
| $\sigma^2_{RE}$ (Variance of random effects on physician level) | | 0.90** | 0.91** |
| Intra-class correlation (in %) | | 21.57 | 21.58 |
| Log-Like | | -288,587 | -288,480 |
| Akaike Info Criterion (AIC) | | 577,216 | 577,006 |
| R²-MacFadden (in %) | | 12.56 | 12.60 |

The first column presents sample means. The other columns display the estimated regression coefficients. Based on 476,260 observations (315,752 patients from 2,189 primary care physicians). Estimated by means of Maximum Likelihood. Significance levels: * 5%, ** 1%.

The results of two logistic regression models with specified random effects on the physician level are shown in the third and fourth column of Table 1. In both models, the estimated intra-class coefficients (21.6%) suggest that conditional on the covariates, almost one quarter of total variation in antibiotic prescription could be explained by the individual physician's practice style. The estimated regression coefficients indicated that patients aged over 35 years were significantly more likely to receive an antibiotic prescription than younger patients. The strongest effect was achieved for patients aged between 35–65 years. Female patients were also more likely to receive an antibiotic. The interaction effects between the patients' gender and age groups in Model (2) signified that the gender difference only exists for patients younger than 35 years. As indicated by the smaller Akaike Information Criterion (AIC), the fit of the model was significantly improved, leading to our final model that was considered for the intervention analysis.

## Matching

The entropy balancing was applied to match physicians of the intervention group with physicians of the control group in the pre-intervention period. In addition to the control variables, the pre-intervention prescription rates served as conditional variables used in the matching. Case- and patient-level variables were aggregated on the physician level. Table 2 shows the means of the variables for the intervention as well as the matched and non-matched control group. Further, the differences between intervention group and unmatched controls as well as the share of missing observations are shown for each variable.

The intervention group is characterized by higher average prescriptions per physician in comparison with the control group. This hints for a selection of the participants in the intervention group due to their pre-intervention outcome. Furthermore, the change over time differed between both groups, underlining that the assumption of parallel trends might not hold. The average number of patients was higher in the control group. However, none of the differences were significant, except for the fraction of patients with a certain diagnosis and family insurance. This might have been due to the low number of observations at the physician level in the intervention group (*n* = 17). Nevertheless, after applying the reweighting approach based on entropy balancing the means in the control group equaled the means in the intervention group.

## Univariate DiD analysis

To assess the sensitivity of the DiD analysis due to model specifications and the balancing we started presenting univariate DiD estimates (simple mean comparison) based on unmatched

**Table 2. Means of aggregated variables before intervention.**

| variables | Intervention group | Control group | | Difference between (a) and (b) | Share of missing observations (in %) |
|---|---|---|---|---|---|
| | | Un matched | matched | | |
| | (a) | (b) | (c) | | |
| *Outcome*: *Prescription rate (in %)* | | | | | |
| 2013 | 51.5 | 46.5 | 51.5 | 5.0 | 3.74 |
| 2014 | 48.3 | 44.7 | 48.3 | 3.6 | 2.01 |
| 2015 | 47.6 | 43.2 | 47.6 | 4.4 | 2.64 |
| *Number of URTI-patients* | | | | | |
| 2013 | 66.5 | 88.4 | 66.4 | -21.9 | 3.74 |
| 2014 | 65.1 | 79.0 | 65.0 | -13.9 | 2.01 |
| 2015 | 69.3 | 82.3 | 69.3 | -13.0 | 2.64 |
| *Share of cases (in %)* | | | | | |
| quarter | | | | | |
| 2nd quarter | 22.3 | 21.8 | 22.3 | 0.5 | 0 |
| 3rd quarter | 18.6 | 18.1 | 18.6 | 0.5 | 0 |
| 4th quarter | 27.6 | 26.2 | 27.6 | 1.4 | 0 |
| (Reference: 1st quarter) | | | | | |
| diagnosis | | | | | |
| sinusitis (J01) | 14.8 | 19.7 | 14.8 | -4.9 | 0 |
| pharyngitis (J02) | 47.2 | 41.3 | 47.2 | 5.9 | 0 |
| (Reference: bronchitis (J20)) | | | | | |
| certainty | | | | | |
| certain diagnosis | 99.3 | 97.9 | 99.3 | 1.4** | 0 |
| service-type | | | | | |
| Emergency services | 6.6 | 5.5 | 6.6 | 1.1 | 0 |
| Patient demographics | | | | | |
| Patients aged 35–65 | 55.0 | 51.5 | 55.0 | 3.5 | 0 |
| Patients aged >65 | 13.6 | 14.0 | 13.6 | -0.4 | 0 |
| Female patients | 59.4 | 60.6 | 59.4 | -1.2 | 0.02 |
| sex-age interactions | | | | | |
| Female patients aged 35–65 | 32.9 | 31.3 | 32.9 | 1.6 | 0.02 |
| Female patients aged >65 | 7.9 | 8.5 | 7.9 | -0.6 | 0.02 |
| Insurance status | | | | | |
| Patients family insurance | 8.7 | 12.4 | 8.7 | -3.7* | 0 |
| Patients pensioners insurance | 15.6 | 16.1 | 15.6 | -0.5 | 0 |
| *Physician characteristics* | | | | | |
| PCP specialty | | | | | |
| General Internists (in %) | 23.5 | 26.4 | 23.5 | -2.9 | 0 |
| (Reference: GP) | | | | | |
| Female physician (in %) | 23.5 | 37.9 | 23.5 | -14.4 | 0 |
| (Reference: male) | | | | | |
| Physician age | 54.3 | 53.6 | 54.3 | 0.7 | 0.00 |
| *Number of PCPs* | 17 | 1,460 | 17 | | |

The first three columns present means of selected variables used for the matching before intervention for trained controls and matched controls, respectively. The last column displays the differences between intervention and control group before matching. Significance levels: * 5%, ** 1%. Patient variables are aggregated on physician level by summing up (*Number of URTI-patients*) or computing as shares of cases.

**Table 3. Univariate difference-in-difference analysis of the communication training on the antibiotic prescribing behavior.**

| | Prescribing rate (in %) | | | Difference-in-Difference | |
|---|---|---|---|---|---|
| | **Intervention Group** | **Control Group** | | | |
| | | **unmatched** | **matched** | **unmatched** | **matched** |
| Before | 55.43 | 47.27 | 52.86 | | |
| *(2014–2015)* | *(n = 2284)* | *(n = 235355)* | *(∑wᵢ≈2282)* | | |
| After | 44.27 | 42.61 | 47.80 | | |
| *(2016)* | *(n = 698)* | *(n = 75167)* | *(∑wᵢ≈736)* | | |
| Difference | -11.16** | -4.65** | -5.07* | -6.51** | -6.10* |
| *p-value* | <0.001 | <0.001 | 0.017 | 0.003 | 0.043 |

313,504 observations (234,723 patients from 1,477 general practitioners). Significance levels: * 5%, ** 1%. Matching is based on Entropy balancing using the variables listed in Table 2 and $w_i$ denotes the Entropy balancing weights.

and matched sample data in Table 3. Neglecting physician-specific effects and other covariates, the reduction in the overall prescription rate of the intervention group between the pre-intervention and post-intervention period was 11.2%. For the control group a reduction of 4.7% could be detected. The difference between both groups in the difference between the periods is the DiD estimator, which is -6.5% [95% CI: (-10.7%; -2.3%)], and significant. Reweighting the observations of the case-level by the entropy weights on the physician-level increased the prescription rate of the matched control group to 52.9%. This was also slightly smaller than the rate of the intervention group (55.4%), which might be, because the matching was done at the physician level and not at the case level. However, the DiD estimate for the matched sample was rather similar (-6.1% [95% CI: (-12.0%; -0.2%)],) and also significant. Concluding, both univariate DiD estimates suggest a significant reduction of antibiotic prescriptions after the communication training.

## Multilevel DiD regression analysis

To take into account the control variables and the random effects on the physician-level, we estimated the specification of Model (2) based on the extended data set, as well as the *DiD* and *training* dummy variable. Table 4 shows the estimated DiD effects and the moderation effects. To ease the interpretation of the estimated DiD coefficient, the marginal effect on the prescription rate was also shown for the direct effects.

All specifications obtained a significant reduction of the prescription rate due to the intervention. There were no substantial differences between the estimates of the matched and unmatched sample. The marginal effects were close to the estimated univariate DiD effects.

The results of a moderation effect of the DiD effect by the age and sex of the patients are also shown in Table 4. They suggest that the intervention had a significantly stronger effect on the treatment of female patients aged below 35.

## Discussion

In this study, we estimated the effect and its moderations of a communication training based on the MAAS-Global-D instrument on the antibiotic prescription rate of primary care physicians for the treatment of upper respiratory tract infections. Since the control group was formed from observational data, we applied a combination of difference-in-difference estimation and statistical matching based on entropy balancing to estimate the intervention effect. Relevant variables for the matching were selected after estimating a multi-level logistic regression model for the antibiotic prescribing decision, based on case-, patient- and physician-level

**Table 4. Multilevel logistic regression analysis of the difference-in-difference effect of the communication training on prescribing an antibiotic.**

|  | (3) | (4) | (5) | (6) |
|---|---|---|---|---|
| matching | no | yes | no | yes |
| trained | 0.15 | -0.08 | 0.14 | -0.08 |
| DiD | -0.31** | -0.28* | 0.19 | 0.14 |
| 95%-CI[a] of DiD | [-0.50, -0.12] | [-0.50, -0.05] |  |  |
| ME[b] of DiD (in %) | -6.34** | -6.44* |  |  |
| 95%-CI[a] of ME (in %) | [-10.31–2.37] | [-11.67, -1.22] |  |  |
| Odds Ratio DiD | 0.73** | 0.76* |  |  |
| 95%-CI of OR | [0.61, 0.89] | [0.60, 0.95] |  |  |
| *Interaction effects* |  |  |  |  |
| DiD*Pat age (35–65) |  |  | -0.52 | -0.49* |
| DiD*Pat age (65+) |  |  | -0.33 | -0.26 |
| DiD*Fem pat (<35) |  |  | -0.73* | -0.65** |
| DiD*Fem pat (35–65) |  |  | -0.13 | -0.03 |
| DiD*Fem pat (65+) |  |  | -0.00 | 0.07 |

313,504 observations (234,723 patients from 1,477 general practitioners). Significance levels: ** 5%, *** 1%.

[a] Confidence interval

[b] Marginal Effect. Estimated coefficients of the control variables and the variance of the random effects are not shown. Matching is based on Entropy balancing using the variables listed in Table 2.

data of the pre-intervention period in the first stage. In the second stage, the same model was estimated, based on matched data and extended by the intervention period and DiD specification.

## Pre-intervention analysis

During the pre-intervention period, an antibiotic was prescribed in almost half of the considered cases. This relatively high number of antibiotic prescriptions is also observed in related studies [57, 59]. In both groups (intervention and control), the prescription rate slightly decreased over time. This is similar to the declining trend of general antibiotic use in other countries [60] and might be explained by an increased awareness of antimicrobial resistance [61], e.g. due to successful antibiotic stewardship programs as the *German Strategy against Antibiotics Resistance* [62]. The estimated intra-class coefficient of the multilevel regression model shows that the individual physician's practice style explains about 22% of the total variance in antibiotic prescription and is similar to the results of related studies [25, 35, 63]. It suggests the prospect of a successful reduction in the prescription rate by changing the individual physician's prescribing behavior. Most of the observable characteristics of the physician do not explain the variance in prescribing behavior. Only the number of URTI patients (serving as a proxy of the physician's workload) increases the probability of antibiotic prescription. This effect underlines the hypothesis that insufficient communication determines the antibiotic prescription. It is more complicated for physicians lacking sufficient time for the consultation due to an overload of patients, to change the patients' expectations [28, 64]. A similar mechanism might explain the positive association of emergency service and the antibiotic prescription probability. In Germany, PCPs face an overload of patients, especially when providing out-of-hours care in emergency service [65]. Patients visiting the emergency service for respiratory complaints might be more severe and therefore might have a strong expectation of receiving an antibiotic [66]. The expectations might also differ between patients, as suggested by the estimated effects of the patient characteristics.

Patients above the age of 35 years receive a significantly higher number of antibiotic prescriptions than younger patients do. For patients belonging to a higher-risk group (e.g., elderly patients) respective guidelines suggest the use of antibiotics in some cases [44–46]. Therefore, the application of guidelines cannot explain the lower prescription rates for patients aged above 65 in comparison with patients aged between 35 and 65 (Model (1): 0.28 vs. 0.33). Differences between the patients' expectations in the age-groups below and above 35 respectively, might be more likely to serve as an explanation. Work pressure and other related stress cause patients to desire rapid relieve from symptoms and cure of their sickness [67]. The perceived importance of the patient's job promotes the decision to prescribe an antibiotic [30, 68]. This might explain that the antibiotic prescription rate considerably exceeds the clinically justified amount for young and middle-aged adults with respiratory infections in the UK [69].

Our results further suggest that women receive more frequent an antibiotic prescription. Women are more likely to visit a physician for URTI than men [57] and are found to be more skeptical towards the physician's suggestions [70]. This patient group might combine higher expectations and wariness that might lead to additional communication requirements. This hypothesis is in line with the higher antibiotic prescription rate that is observed for female patients in our data. However, the underlying mechanisms were not aim of our research focus and is, therefore, subjected to future research. Similar to other studies, the gender gap vanishes with the increasing patient-age [71]. Only female patients below the age of 35 receive a significantly higher number of antibiotic prescriptions. This result might indicate that the communication problem is mostly pronounced for the treatment of this group of patients. In the following, we discuss the effects of the communication training on the antibiotic prescription probability.

### Intervention analysis

We applied different approaches and specifications to robustly estimate the effect of the communication training on the antibiotic prescribing behavior. The univariate approach estimates an 11-percentage-point reduction of prescriptions for the intervention group after the training. This result is very similar to a related study [72]. All our approaches (univariate and multivariate) estimate a decrease of around 6.5 percentage-points in the prescription probability of the trained physicians. These robust estimates are in line with the findings of other related studies applying RCT methodology [32, 35, 73]. The moderation analysis confirms that the effect of the communication training is stronger for the treatment of patients marked by a larger communication problem. The impact of the training on the reduction of antibiotic prescription is significantly stronger for the treatment of young women. Thus, physicians with improved communication skills might be able to better address the potentially higher expectations of young female patients to receive an antibiotic therapy and their wariness towards the physician's suggestions [70].

As argued by Fritz and Holton [74], the lack of trust in the patient-doctor relationship enhances the likelihood of overprescribing. A patient trusting in the physician's clinical judgment, can be reassured to accept non-prescribing [75]. Furthermore, secured trust between the patient and physician could reduce the probability of the physician to misperceive the patient's expectation to receive antibiotic treatment. To establish a trustful relationship it is important for the patient to recognize the physician's trust in them and believe that the physician acts in their best interest [76]. Signals of trustworthiness are given by verbal and nonverbal communication and serve to establish patients' trust, and, thus, influence the doctor-patient relationship [77]. For this purpose, the MAAS-Global-D might be a promising tool to improve effective communication since both verbal and nonverbal communication skills are

part of the training. To comprehend emotions as well as feelings and to react adequately, the MAAS-global-D-manual proposes the physician to render the feelings expressed by the patient during the consultation either in words or nonverbally [39]. Trust is considered for most patients to be an integral part of an ongoing relationship with a physician [78]. An increased continuity of care enables, on the one hand, physicians to better evaluate the patient's expectations of receiving an antibiotic by the more intimate knowledge of their living conditions. On the other hand, patients can build up a deep understanding of appropriate antibiotic use and will change their expectations permanently. The findings of Robert et al. [79] suggest that receiving information about antibiotics from family physicians is usually not associated with an increased knowledge of the patients. A trustful and continued relationship might be helpful for physicians to provide information about the use of antibiotics, and to improve knowledge about antibiotics especially among target groups [79, 80]. As we found in our pre-intervention data analysis, one specific target group consists of young female patients.

## Limitations and strengths

The study estimated the effects of a communication training for primary care physicians on the antibiotic prescription rate for infections of the upper respiratory tract and its moderation by age and gender of the patients.

The study has strengths as well as limitations. In contrast to the previously planned randomized controlled trial [47], in this study we formed a control group from observational data. In contrast to the control group physicians the members of the intervention group did know that their prescription data would be analyzed for the periods before and after the training. Therefore, we cannot exclude that behavior change in the intervention group is due to the physicians' awareness of being under observation rather than solely due to the intervention (Hawthorne effect) [81]. However, since we considered the data of the physicians up to one year after the training, we do not believe that this effect is responsible for persistent behavior changes. Further, the approach that has been applied to estimate the intervention effect is more sophisticated and is, thus, more susceptible to misspecification than an RCT [82]. To minimize the risk of biased estimates we applied several alternative approaches (univariate, multivariate, matching, no-matching) and specifications (fixed and random effects) as robustness checks. All estimated effects of the intervention are very similar. Therefore, we believe that misspecification is not a big issue here.

A strength of this study is that it relies on routine data collected from all primary care physicians in a specific region of Germany. The relatively large number of physicians of the (matched) control group (n = 1,460) might ensure a higher external validity of our findings than the rather small sample sizes of other related studies applying an RCT [32, 33, 35, 36]. However, the small number of the intervention group highlights the problem to convince PCPs to participate in intervention studies [83, 84]. Another reason for the low response rate might have been rooted in the PCPs' (who already faced an overload of patients) concerns that improved communication skills would prolong the consultation, although so far, there is no evidence to support this claim [85].

While on the one hand, the focus on the federal state of Schleswig-Holstein constrained the representativeness of the findings, it on the other hand also reduced practice variations based on regional differences and state-specific regulations [86]. The analyzed moderation of the patient's age and gender on the communication training effect further increased the insights of antibiotic prescribing behavior. In line with the findings of our pre-intervention data analysis, our results suggest that improved communication skills are mostly effective in cases where the underlying communication problem is particularly pronounced due to high expectations of

the patient to receive an antibiotic or due to the physicians' perceptions. To clarify the moderating role of expectation and its perception for the communication training effect on antibiotic usage future research should include direct measures of these variables [25].

## Conclusion

In this study, we estimated the effect and its moderations of a communication training on the antibiotic prescription rate of primary care physicians for the treatment of upper respiratory tract infections, i.e. acute bronchitis, sinusitis and pharyngitis. The short communication training has been based on the MAAS-Global-D [40], the German version of the Dutch instrument MAAS-Global [39]. Since the control group has been formed from observational data, a combination of difference-in-difference (DiD) and matching has been applied to estimate the intervention effect. To minimize the risk of biased estimates we applied several alternative approaches and specifications as robustness checks that all reveal similar intervention effects. The results show that the communication training decreases the prescribing probability by around 6.5-percentage-points for the physicians of the intervention group. For the treatment of female patients aged below 35, the intervention has a stronger impact.

Our results suggest that communication skills implemented via MAAS-Global-D-training lead to more prudent prescribing of antibiotics for URTIs. Therefore, the MAAS-Global-D-training could not only avoid unnecessary side effects but could also help to reduce the emergence of drug resistant bacteria. The instrument MAAS-Global-D has been proven to provide a valid tool for a training of physicians that encourages an effective communication with the patient. In the Netherlands, communication training is an integral part in the postgraduate-training program of general practitioners. A similar communication training based on the MAAS-Global-D could also be applied in Germany, as well as in other countries, where postgraduate training schemes of PCPs lack in training of communication skills. The instrument and the explanatory manual in German language are available for free download [87].

## Trial registration

The intervention and the previously planned randomized controlled trial (RCT) has been registered in the German Clinical Trial Register (DRKS00009566).

## Supporting information

**S1 Checklist. STROBE statement—checklist of items that should be included in reports of observational studies.**
(DOCX)

**S1 Data. STATA software codes.**
(ZIP)

## Author Contributions

**Conceptualization:** Christoph Strumann, Jost Steinhaeuser, Andreas Sönnichsen, Katja Goetz.

**Data curation:** Timo Emcke.

**Formal analysis:** Christoph Strumann.

**Funding acquisition:** Jost Steinhaeuser, Andreas Sönnichsen.

**Investigation:** Christoph Strumann, Jost Steinhaeuser, Katja Goetz.

**Methodology:** Christoph Strumann, Katja Goetz.

**Project administration:** Katja Goetz.

**Resources:** Jost Steinhaeuser, Katja Goetz.

**Software:** Christoph Strumann.

**Supervision:** Katja Goetz.

**Validation:** Christoph Strumann, Jost Steinhaeuser, Timo Emcke, Andreas Sönnichsen, Katja Goetz.

**Writing – original draft:** Christoph Strumann.

**Writing – review & editing:** Christoph Strumann, Jost Steinhaeuser, Timo Emcke, Andreas Sönnichsen, Katja Goetz.

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
