## [Decision Letter · Decision Letter 0]

25 Feb 2020

PONE-D-19-35281

Communication training and the prescribing pattern of antibiotic prescription in primary health care

PLOS ONE

Dear Dr. Strumann,

Thank you for submitting your manuscript to PLOS ONE. After careful consideration, we feel that it has merit but does not fully meet PLOS ONE’s publication criteria as it currently stands. Therefore, we invite you to submit a revised version of the manuscript that addresses the points raised during the review process.

One of the previous reviewer is happy with the answers provided.

The second previous reviewer was not available for this re-submission and a third reviewer has raised some additional minor concerns. Please address them.

We would appreciate receiving your revised manuscript by Apr 10 2020 11:59PM. To enhance the reproducibility of your results, we recommend that if applicable you deposit your laboratory protocols in protocols.io, where a protocol can be assigned its own identifier (DOI) such that it can be cited independently in the future. For instructions see: http://journals.plos.org/plosone/s/submission-guidelines#loc-laboratory-protocols

We look forward to receiving your revised manuscript.

Kind regards,

Martin Chalumeau, MD-PhD

Academic Editor

PLOS ONE

Journal Requirements:

"CS, KG, TE and JS: the authors declare that they have no competing interests.

I have read the journal's policy and the author AS of this manuscript have the following competing interests: AS is an editor of the German version of EbMG. The online access is provided free of charge to the study PCPs by Verlagshaus der Aerzte, Vienna."

4. Your ethics statement must appear in the Methods section of your manuscript. If your ethics statement is written in any section besides the Methods, please move it to the Methods section and delete it from any other section. Please also ensure that your ethics statement is included in your manuscript, as the ethics section of your online submission will not be published alongside your manuscript.

Reviewers' comments:

Reviewer's Responses to Questions

**Comments to the Author**

1. Is the manuscript technically sound, and do the data support the conclusions?

Reviewer #1: Yes

Reviewer #2: Yes

2. Has the statistical analysis been performed appropriately and rigorously? 

Reviewer #1: I Don't Know

Reviewer #2: Yes

3. Have the authors made all data underlying the findings in their manuscript fully available?

Reviewer #1: Yes

Reviewer #2: No

4. Is the manuscript presented in an intelligible fashion and written in standard English?

Reviewer #1: Yes

Reviewer #2: Yes

5. Review Comments to the Author

Reviewer #1: This is a well-written and clear manuscript that discusses an important antibiotic stewardship intervention with potential implications for both public health and clinicians which would be a helpful addition to the literature. Despite the small intervention sample size, I do think it is worth publication, and so have some suggestions for the authors to improve this manuscript. Additionally, I have very little experience with entropy balancing and using observational data as a control and, therefore, would recommend the journal consider a statistical review to ensure the approaches used were appropriate as I am unable to provide a full evaluation.

Suggestions:

• Throughout the paper, the authors reference inappropriate antibiotic prescribing. However, with the diagnoses listed here, there are some cases where antibiotic prescribing is appropriate (at least based on the information provided), so it is too speculative to call all prescribing for these conditions inappropriate. The authors either need to present more data about what proportion of these diagnoses might warrant antibiotic therapy (I am not sure if this data is available in the literature) or be very cautious and explicit in their interpretations.

• Introduction: Overall, a clear and concise summary of the evidence. At times, statements summarizing the literature seem over-generalized or too broad. This is especially pronounced on pages 3-4, starting with the sentence that begins with “Most patients possess insufficient knowledge…”. I believe all that is needed to ameliorate this is softening the language, for example, consider changing “Additionally, physicians wrongly assume to “Additionally, physicians may wrongly assume….”.

• Introduction: I found the wording in the last paragraph of the introduction to be slightly confusing. Consider rephrasing and more succinctly stating the aims of the study.

• Methods: How were antibiotic prescriptions linked to diagnoses? Are indications listed for the prescription or was it inferred based on the visit diagnoses? If the latter, were visits with diagnoses for conditions where antibiotics are appropriate excluded (e.g., UTI, pneumonia)?

• Methods: I saw in the response to previous reviewer comments that tried to address the certainty of diagnosis variable, but I am still confused by this. It might be helpful to include more information about how this is recorded in the claim/medical record.

• Methods: were there any collinearity issues in your model when using both age and insurance states as these reflect similar “stage of life” clusters?

• Tables: I found the tables header in table 1 and 4 difficult to follow. I think it would be clearer to put a more descriptive title than just the model number, or if that is not feasible, include in a footnote the model specifics. This information is in the methods, but the tables need to be interpretable on their own.

• Discussion: I think there might be other reasons for a declining trend in antibiotic use besides awareness of AMR. Consider other secular changes such as visit and disease incidence, antibiotic stewardship practices, and immunizations.

• Discussion, page 16: The points about emergency medicine visits are fair, but the authors should also consider that these visits may be more severe than scheduled primary care visits.

• I found the discussion of why younger adults and women might receive higher antibiotics speculative and unconvincing. Consider revising.

• Limitations: regarding the Hawthorne effect, did intervention clinicians know their data was being tracked? Were they aware of how long their data would be tracked? These are important considerations for understanding the extent of possible bias due to the Hawthorne effect.

Reviewer #2: Manuscript ID PONE-D-19-15846: Communication training and the prescribing pattern of antibiotic prescription in primary health care: A case-control study.

This manuscript reports the effect of a communication training intervention on the antibiotic prescription rate of primary care physicians (PCP) for the treatment of upper respiratory tract infections in adults.

The information reported in this manuscript is interesting and the statistical approach seem appropriate, on a subject which remains a major public health concern worldwide. However several points should be considered.

A) Major comments:

1) You state that your methodology (Difference-in-Difference estimation and matching approach) allows the demonstration of a causal relationship between the intervention and the outcomes. Such quasi-experimental designs seem to be a valuable option in numerous situations where the classic controlled randomized design is not feasible, and is clearly stronger than observational designs such as before-after studies. However, demonstrating a causal relationship remains particularly challenging, and statistical analysis comparing results from some RCTs to such DID finding similar results are probably not a sufficient proof (your reference 52). Even such RCTs often provide discordant conclusions, and sometimes only meta-analysis of well conducted RCTs allow providing the demonstration of a causal effect of an intervention. Moreover, to my knowledge, it is not recognized by international guidelines of level of evidence gradation, such as the GRADE system, that DID with matching is comparable to RCTs in term of level of evidence. Could you please provide clear support of this? If not, I strongly support removing this affirmation from all the manuscript (actually you used 8 times the term “causal” and 2 times “demonstrate”), including the abstract.

Author Response: Thank you very much for your hint. You are right; we stated too loosely

that we demonstrated a causal relationship. We applied the combination of DID

with matching and several robustness checks to minimize the risk of biased

estimates. However, of course, we cannot exclude that the results are biased with

some accepted statistical error or even examine the risk that might be left. We

therefore revised the manuscript accordingly to your suggestion by deleting the

term “causal” and “demonstrate” when we are referencing to our estimated effects.

Reviewer response: The authors adequately took into account the reviewer suggestion.

2) Your first hypothesis H1 is that your intervention could reduce inappropriate antibiotic prescriptions. However, all you analyses are focused to the reduction of overall antibiotic prescription rates, without detailing which prescriptions are appropriate or not. Moreover, you did not detail German guidelines on antibiotic treatments for URTIs in adults, to allow distinguishing appropriate and inappropriate antibiotic prescriptions. And finally in the discussion you stated that this hypothesis H1 is supported by your results.

I suggest to provide detailed data and analysis on which prescriptions were appropriate or not, in term of indication, class of molecule, duration of treatment, etc; or to change this hypothesis by “reducing overall prescription rate for URTIs”, which was the objective that you previously published in your protocol (your reference 50).

Author Response: Thank you very much for this important comment. In the previous

version of the manuscript, we have formulated our hypothesis too broadly.

Unfortunately, our data set only provides the information about the diagnosis and

whether an antibiotic has been prescribed. Therefore, we cannot distinguish

between appropriate and inappropriate antibiotic prescriptions. We have implicitly

argued that in the case of treating upper respiratory tract infections (URTIs) any

antibiotic prescription is considered to be inappropriate. However, this is not true,

as we have written in the manuscript “For patients belonging to a higher-risk group

(e.g., elderly patients) respective guidelines suggest the use of antibiotics in some

cases.” Moreover, in the revised version of our manuscript we describe in more

detail the cases, where respective guidelines suggest the use of antibiotics (as

suggested by reviewer 2). Therefore, we very welcome your suggestion to change

our hypothesis to “reducing overall prescription rate for URTIs”. Since we have

transferred and adapted the parts with the development of the hypotheses to the

discussion, as suggested by the second reviewer, we have excluded the

hypotheses from the text. However, we scrutiny adapted any misleading wording

in regard to your suggestion that we are aiming to reduce the overall prescription

rate for URTIs by means of the communication training.

Reviewer response: The authors adequately took into account the reviewer suggestion., no additional change needed.

3) Your initial aim was to conduct a randomized trial, but you stated that, due to lower inclusion rate than expected, and a lack of power, the analysis based on this trial was not statistically significant. Then, you reported that you “formed a control group from observational data” to apply a DID method and your matching.

Your initial number of eligible PCP was 1554, but only 17 received the intervention. I don’t understand how many were allocated in the control group initially planned? None of them? The initial planned RCT had 3 arms, how much were in the third arm? Finally in the results section, you talk about 2189 eligible CPC? It is very difficult to follow it without any clear flow chart. If there was also 17 CPC in the initial control group, I don’t understand your matching, which was 1:1 if I well understood (It is not specifically mentioned) could increase your sample size? Finally, the total number of patients involved in the study is not clearly provided. A flow chart with the number of CPC and the number of patients at each stage would be very helpful.

In the same way in the abstract, you said that 1554 PCP were invited to participate, but you neither report the number of PCP finally recruited, nor the number of patients.

Author Response: Our study consists of two different analyses. In the first part, we conduct a preintervention

analysis based on data from 2013 to 2015 of 2,189 PCPs. This is the number of enrolled physicians from 2013 to 2015 with non-missing data. In the second part of the

study, we carried out the intervention analysis, which included data from 1,477 PCPs.

In the study protocol, the sample size has been computed to be 31 per study arm. In total,

we have been able to recruited 34 physicians to participate in the study. Since the inclusion

rate was much lower, both intervention groups have been consolidated. Splitting the group

of 34 physicians resulted in 17 physicians that have been treated and 17 physicians that

have assigned to the control group. However, due to the small number of recruited

physicians even a comparison between the pooled intervention group and the control group

had not resulted in statistical significant effects. That is the reason why we alternatively

formed the control group from observational data consisting of 1,460 PCPs.

Since, we expect that the evolutions of the prescription rates are suspected to differ

between the intervention and control group we apply a matching approach that is based

on covariates and pre-intervention outcomes. The covariates are the same that have been

used in the pre-intervention analysis. The additional conditioning on pre-intervention

outcomes enables that all potential outcome trends are aligned between the intervention

and control group.

In the matching approach, we do not apply a 1:1 matching. Instead, we extracted weights

for each physician of the control group such that the control and the intervention group are

balanced accordingly to pre-intervention out-comes and covariates. Physicians that are

more similar to the trained physicians of the intervention group have higher weights.

Therefore, we use all of the 1,460 PCPs of the control group; however, the specific PCPs

are weighted accordingly to their entropy weights. The estimation procedure takes into

account the weighting of the physicians and we did not just blow up the sample size.

We are aware about showing the results of both analyses (pre- and intervention) is much

material for one manuscript. However, we prefer to show also the results of the preintervention

analysis to increase the transparency. To ease the understanding we provide

in the new version of the manuscript a flow chart with the number of PCP and the number

of treated URT cases at each stage.

Reviewer comment: The number of PCP and patients at eauch stage of the study appear clearer thanks to the flow chart. No additional change needed.

Other comments:

1) Could you define more precisely the study periods for each group? In the published protocol of the initial RCT, you planned to only include months from April to June, from 2013 to 2016. Was it the case in this study? If it is, could explain this choice? It is well known that most of the URTI diagnoses and prescriptions occur in the winter period.

Author Response: Since in the first quarter of 2016 the training has been conducted, the

periods initially planned to include in the study has been restricted to April to June.

We could increase the sample length until the end of 2016. However, the first period

of 2016 is still excluded from the study due to the training period.

Reviewer response: we understand the author choice regarding the inclusion period, not additional change required.

2) Your main analysis report a significant reduction of antibiotic prescriptions, without detailing for which pathology. Do you have some precision about it? One concern when reducing antibiotic prescriptions is to increase the risk of treatment failure/complications. Do you have any data about hospitalization rate before and after the intervention, or the number of readmission?

Author Response: No, unfortunately the data do not include any further information, e.g.

about hospitalization. We only know the target-diagnoses of acute bronchitis,

sinusitis and pharyngitis of the specific patient treated.

Reviewer response : we understand the author response, however the first part of the question has no answer. It would be interesting to know which pathology beneficiate from the higher reduction of ATB use thanks to the intervention.

3) Another concern of such intervention is the long-term benefice. If I well understood, you assessed the effect of the intervention only few months after. Could you justify this choice? An important cluster randomized trials on antimicrobial stewardship showed that such intervention could have a transient effect (doi: 10.1001/jama.2014.14042), and without any continued feedback to clinician, the durability of the intervention could be jeopardized. Could you discuss it in the limits? Did you plan to provide feedback, which is an important component of antimicrobial stewardship?

Author Response: This is a very good idea and of course an important concern for the

limitations that we have now included in the new version of the manuscript. Thank

you very much! Unfortunately, our choice is based on our data set. We only have

data until the end of 2016. For future research, this is an interesting question.

Reviewer response: no additional comment.

4) Could you detail why among 1554 invited PCP, only 17 received the intervention? This is less than 1.5%, and could represent a substantial risk of selection bias, since the PCP who accepted to participate may be particularly aware about the need to reduce antibiotic use, while the remaining PCP, which constituted your control observational data, may be less interested in this. Could you discuss it in the limits? Again, this should lead you to be very careful in using the “causality” term, because this characteristic cannot be recorded in the baseline variables, and thus is very difficult, if not impossible, to account for in the matching analysis.

Aurhor Response: The risk of selection bias is exactly the reason why we applied the matching

approach that is based on pre-intervention outcomes. After weighting the observations of

the control group by the physician specific entropy weights, the outcome trends between

the intervention and control group are similar as shown in Table 2. This means that

physicians that have a similar prescribing behavior over time before the intervention have

higher weights. Of course, we cannot guarantee that the applied matching takes into

account the distinct awareness between the PCPs who accepted to participate and the

ones who do not accepted. Therefore, we deleted the term “causal” and “demonstrate”

when we are referencing to our estimated effects and add this issue to the limitations.

Reviewer response: we agree with the author response and changes in the manuscript.

5) Could you explain why you choose only 3 CIM10 codes for URTIs, is there no other relevant code?

Author Response: We concentrate the analysis to these diagnoses, since in the primary care

setting for these cases an antibiotic is often prescribed, although only in some cases the

use of antibiotics is suggested by respective guidelines within these diagnoses.

Reviewer response: We understand the author respons, but could you at least provide the proportion represented by these three codes among all the ARTIs ? This would be helpful to be confortable to extrapolate your findings to any ARTIs

6) Your main analysis report a significant reduction of 6.5% of antibiotic prescriptions compared to the control group. Due to the major burden of antibiotic resistance,; even this small reduction may have an impact, but could you discuss, from an economic point of view, why you believe that this intervention may be more cost effective than another to reduce antibiotic use in the community?

Author Response: We have trained primary care physicians to improve their communication skills,

since literature suggests, that antibiotic prescriptions can be associated with a

communication problem. Our results underline this hypothesis. Other studies have

implemented different interventions like providing information material about antibiotic use

for the patients or integrated a warning button in the physicians’ software. To assess the

cost effectiveness of these interventions one needed to compare the effects and the

underlying costs that are hardly available. However, we believe our intervention is cost

effective because of two reasons. First, our intervention consists of a short (2 x 2.25h)

communication training that can be easily integrated in advanced clinical education, as it

is already part of under- and postgraduate training in the federal state of Schleswig-

Holstein. Second, we believe that the effects of the communication training are not limited

to the antibiotic prescriptions behavior. In future research, we plan to analyze the training

effect on other behavioral aspects of the physicians, for instance withdrawals of protonpump

inhibitors prescriptions.

Revier response: no additional comment.

6. PLOS authors have the option to publish the peer review history of their article (what does this mean?). If published, this will include your full peer review and any attached files.

Reviewer #1: No

Reviewer #2: Yes: Naïm Ouldali

---

## [Author Response · Author response to Decision Letter 0]

7 Apr 2020

Response to Reviewers, PLOS ONE, Submission PONE-D-19-35281

Communication training and the prescribing pattern of antibiotic prescription in primary health care 

Thank you very much for the review of our paper. Again, it was very helpful indeed for improving the previous version of the manuscript. We incorporated (almost) all suggestions. Below you will find the specific responses to the reviewer’s comments.

Review Comments to the Author

Reviewer #1

This is a well-written and clear manuscript that discusses an important antibiotic stewardship intervention with potential implications for both public health and clinicians which would be a helpful addition to the literature. Despite the small intervention sample size, I do think it is worth publication, and so have some suggestions for the authors to improve this manuscript. Additionally, I have very little experience with entropy balancing and using observational data as a control and, therefore, would recommend the journal consider a statistical review to ensure the approaches used were appropriate as I am unable to provide a full evaluation.

Suggestions:

• Throughout the paper, the authors reference inappropriate antibiotic prescribing. However, with the diagnoses listed here, there are some cases where antibiotic prescribing is appropriate (at least based on the information provided), so it is too speculative to call all prescribing for these conditions inappropriate. The authors either need to present more data about what proportion of these diagnoses might warrant antibiotic therapy (I am not sure if this data is available in the literature) or be very cautious and explicit in their interpretations.

Response: Thank you very much for this important comment. Of course, you are right; there are some cases where antibiotic prescribing is appropriate in cases of the listed diagnoses. We have described these cases in the Data source subsection. Unfortunately, our data set do not provide additional information that would enable us to assess the appropriateness of an antibiotic prescription. To make this point more explicit in the revised version of our manuscript, we indicate the aim of the study as to reduce the overall prescription rate for URTIs by means of the communication training. The term “inappropriate prescription” has been deleted to avoid any misleading wording. 

• Introduction: Overall, a clear and concise summary of the evidence. At times, statements summarizing the literature seem over-generalized or too broad. This is especially pronounced on pages 3-4, starting with the sentence that begins with “Most patients possess insufficient knowledge…”. I believe all that is needed to ameliorate this is softening the language, for example, consider changing “Additionally, physicians wrongly assume to “Additionally, physicians may wrongly assume….”.

Response: At several stages, we have softened the language by inserting words as “may”, “might” etc.

• Introduction: I found the wording in the last paragraph of the introduction to be slightly confusing. Consider rephrasing and more succinctly stating the aims of the study.

Response: In the revised version of the manuscript, the aim of the study is stated more compactly.

• Methods: How were antibiotic prescriptions linked to diagnoses? Are indications listed for the prescription or was it inferred based on the visit diagnoses? If the latter, were visits with diagnoses for conditions where antibiotics are appropriate excluded (e.g., UTI, pneumonia)?

Response: Thank you very much for this important hint that we had not addressed in the previous version of the manuscript. We now have added the following paragraph in the Data source subsection. “Since the antibiotic prescriptions have been inferred based on the visit diagnoses, we excluded cases with additional diagnoses. This includes the presence of diagnoses regarding puerperium/pregnancy (O00-O99), further (bacterial) infections (A00 to A37, A39 to A79, J15, J17, J18) or chronic diseases (I50, J44, J45, C00 to C75). If the diagnosis had been made several times or more than one diagnosis had been made from the three groups (J01, J02, J20), the corresponding cases were also excluded.”

• Methods: I saw in the response to previous reviewer comments that tried to address the certainty of diagnosis variable, but I am still confused by this. It might be helpful to include more information about how this is recorded in the claim/medical record.

Response: We included further information about the German coding policy for primary care physicians regarding the certainty of the diagnosis in the Measurements subsection.

• Methods: were there any collinearity issues in your model when using both age and insurance states as these reflect similar “stage of life” clusters?

Response: Yes, there is an issue regarding collinearity, especially for the variables Patient aged 65+ and Pensioners insured. If we neglect the variable Pensioners insured, the estimated effects of Patient aged 65+, Female patient aged 35-65 and Female patient aged 65+ are slightly stronger. All other effects are not affected. However, in our point of view, this collinearity problem is not critical, since it does not affect our main results. Moreover, we decided to include the variable Pensioners insured, since 3% of the observations are indicated as Pensioners insured, but are below the age of 65 and 1.5% of the observations are not indicated as Pensioners insured that are aged above 65. To summarize, we think that Pensioners insured serves as an important control variable and the collinearity issues can be neglected.

• Tables: I found the tables header in table 1 and 4 difficult to follow. I think it would be clearer to put a more descriptive title than just the model number, or if that is not feasible, include in a footnote the model specifics. This information is in the methods, but the tables need to be interpretable on their own.

Response: We extended the header and notes of the tables.

• Discussion: I think there might be other reasons for a declining trend in antibiotic use besides awareness of AMR. Consider other secular changes such as visit and disease incidence, antibiotic stewardship practices, and immunizations.

Response: Thank you very much for this hint. In the revised version of the manuscript, we extended the discussion by mentioning antibiotic stewardship programs. Since we included the number of URTI patients, the visit and disease incidence might be controlled for and do not serve as an explanation here. 

• Discussion, page 16: The points about emergency medicine visits are fair, but the authors should also consider that these visits may be more severe than scheduled primary care visits.

Response: We integrated this important point in the discussion.

• I found the discussion of why younger adults and women might receive higher antibiotics speculative and unconvincing. Consider revising.

Response: We agree; this part of the discussion is somewhat speculative. However, our aim of the study was to gain further insights about the antibiotic prescribing process. Our hypothesis was that patients’ expectations are a key factor driving inappropriate antibiotic prescription due to a larger underlying communication problem. As we argue by means of the findings of other studies, the expectations might vary with age and gender. The underlying mechanisms behind these factors are far beyond our research focus. Therefore, the discussion remains here speculative. However, using these structural differences in the antibiotic prescribing behavior - potentially reflecting differences in the patients’ expectations - allows us to gain insights about differences in the effects of the communication training in the form that the higher the communication problem the more effective is an communication improvement. This result seems to be trivial, but it obtains insights of the underlying mechanism that is mostly not considered by other related studies.

However, to reduce the extent of the speculation, we revised the respective paragraph accordingly. 

• Limitations: regarding the Hawthorne effect, did intervention clinicians know their data was being tracked? Were they aware of how long their data would be tracked? These are important considerations for understanding the extent of possible bias due to the Hawthorne effect.

Response: Yes, the participants did know that their data would be tracked before and after the intervention. Therefore, of course, we cannot control for the Hawthorne effect. This is an important limitation of our study that we discuss in the limitation section.

Reviewer #2

Manuscript ID PONE-D-19-15846: Communication training and the prescribing pattern of antibiotic prescription in primary health care: A case-control study.

Reviewer response: no additional comment.

---

## [Editor Report · Decision Letter 1]

5 May 2020

Communication training and the prescribing pattern of antibiotic prescription in primary health care

PONE-D-19-35281R1

Dear Dr. Strumann,

We are pleased to inform you that your manuscript has been judged scientifically suitable for publication and will be formally accepted for publication once it complies with all outstanding technical requirements.

With kind regards,

Martin Chalumeau, MD-PhD

Academic Editor

PLOS ONE
---

## [Editor Report · Acceptance letter]

8 May 2020

PONE-D-19-35281R1 

Communication training and the prescribing pattern of antibiotic prescription in primary health care 

Dear Dr. Strumann:

I am pleased to inform you that your manuscript has been deemed suitable for publication in PLOS ONE. Congratulations! Your manuscript is now with our production department. 

With kind regards,

on behalf of

Dr. Martin Chalumeau 

Academic Editor

PLOS ONE